# The care needs of persons with oropharyngeal dysphagia and their informal caregivers: A scoping review

**Aurora Ninfa**[1]*, **Valeria Crispiatico**[1], **Nicole Pizzorni**[2], **Marta Bassi**[2], **Giovanni Casazza**[2], **Antonio Schindler**[2], **Antonella Delle Fave**[1]

1 Department of Pathophysiology and Transplantation, Università degli Studi di Milano, Milan, Italy,
2 Department of Biomedical and Clinical Sciences "Luigi Sacco", Università degli Studi di Milano, Milan, Italy

\* aurora.ninfa@unimi.it

**Data Availability Statement:** All relevant data are within the paper and its Supporting information files.

## Abstract

### Introduction

Besides affecting physical health, Oropharyngeal Dysphagia (OD) entails limitations in daily activities and social participation for both patients and their informal caregivers. The identification of OD-related needs is crucial for designing appropriate person-centered interventions.

### Aims

To explore and map the literature investigating the care needs related to OD management of adult persons with OD and their informal caregivers during the last 20 years.

### Methods

A scoping review was conducted and reported following PRISMA guidelines. Five electronic databases and reference lists of eligible publications were searched for original works in English or Italian, published between January 2000 and February 2021. Two independent raters assessed studies' eligibility and extracted data; a third rater resolved disagreements. Extracted care needs were analyzed using a Best fit framework synthesis approach.

### Results

Out of 2,534 records preliminarily identified, 15 studies were included in the review and 266 care needs were extracted. All studies were conducted in Western countries. Research methods primarily consisted of qualitative interviews and focus groups (14 studies, 93.3%); head and neck cancer was the most frequent cause of patients' dysphagia (8 studies, 53.3%); caregivers' perspective was seldom investigated (5 studies, 33.3%). Both patients and caregivers primarily reported social (N = 77; 28.9%) and practical (N = 67; 25.2%) needs, followed by informational (N = 55; 20.7%) and psychological (N = 54; 20.3%) ones. Only patients reported physical needs (N = 13; 4.9%), while spiritual needs were not cited.

**Funding:** This work was supported by: "Università degli Studi di Milano, Piano di sostegno alla ricerca - Linea 2". The funder had no role in study design, data collection and analysis, decision to publish, or preparation of the manuscript.

**Competing interests:** The authors have declared that no competing interests exist.

## Conclusions

The recurrence of personal and social needs besides physical ones highlighted the manifold impact of OD on patients' and caregivers' lives. Larger and more focused studies are required in order to design tools and interventions tailored to patients' and caregivers' needs.

## Introduction

Oropharyngeal Dysphagia (OD) is a clinical condition that implies abnormalities in the physiology of oropharyngeal swallowing. Swallowing safety (i.e. transfer of the bolus from the mouth to the esophagus without penetration or aspiration into the airway) and/or efficiency (i.e. transfer of the bolus from the mouth to the esophagus without residue) might be impaired. OD may be associated with life-threatening complications, such as aspiration pneumonia, malnutrition and/or dehydration [1–3]. OD prevalence in the general population amounts to 12.1% [4], while it shows ample variations in persons with stroke (8.1–80%), Parkinson's disease (11–81%), traumatic brain injury (27–30%), community-acquired pneumonia (91.7%) [5], and head and neck cancer (HNC; 66% at diagnosis) [6].

Patients with OD may suffer from mild difficulties to severe inability to swallow, requiring bolus modifications and alternative feeding methods (i.e. nasogastric tube, gastrostomy tube) [7,8].

OD severity, diet restrictions to prevent complications, as well as related behavioral and social limitations may lead patients to experience depression, isolation [9], and decreased health-related quality of life (HRQOL) [10,11]. Changes in the meaning and experience associated with eating can also negatively impact on patients' social roles [12].

Family members and friends of persons with OD may also be negatively affected by the condition, since they are frequently involved in the care process as informal caregivers, a term referring to individuals who provide practical and/or emotional support without payment [13]. Regardless of OD etiology, caring for persons with OD may result in remarkable physical and emotional burden [14–20].

OD clinical characteristics and impact on patients' and caregivers' lives were both investigated through primary evidence and synthetized by means of literature reviews. In particular, reviews targeted the management of OD in acute and critical care [21,22] and in nursing homes [23], the production of suitable food using texture modification technologies [24], the impact of OD and modified-texture diets on patients' HRQOL [10,11], and the burden perceived by caregivers [25].

The physical and psycho-social consequences of OD for both patients and their informal caregivers are conducive to emerging needs. In motivational terms, needs refer to "the presence of a particular desire or preference, often rooted in a deficit or shortage, with such preferences varying widely between individuals" [26]. In the medical context, they comprise any necessity arising in the physical, practical, informational, social, psychological, and spiritual domains during the diagnostic, treatment, and follow-up phases of a disease [27], and requiring health professionals' assessment and intervention.

To date, most efforts to address patients' and caregivers' needs consist of frameworks developed to increase efficiency and equity of healthcare services, taking into account organizational and contextual factors hindering or facilitating service access and use. Examples of this approach are the global WHO Framework on integrated people-centered health services

(IPCHS) [28] and other more specific frameworks contextualized in community and palliative care [29–32]. In all these frameworks, however, the unit of analysis is the healthcare service rather than the person. The Supportive Care Framework (SCF) [27] attempts to classify individual needs of patients diagnosed with cancer along the different stages of disease through appropriate and timely services. The needs of persons with cancer were classified into seven domains: physical, emotional, practical, informational, social, psychological, and spiritual. Over the decades, the SCF was fruitfully applied to other clinical conditions such as Amyotrophic Lateral Sclerosis [33] and stroke [34]. To the best of our knowledge, however, it has never been used to investigate and classify the needs of patients with OD and their informal caregivers.

Another relevant aspect yet to be explored among patients with OD and their caregivers is the variety of needs-driven coping responses that shape their behavioral and psychological adaptation to OD. To this purpose, the Common-Sense Model of Self-Regulation (CSM) [35,36] may represent a useful framework. According to the CSM, when facing or anticipating a health threat (illness) individuals develop personal beliefs about both the threat (illness representations) and the strategies required to manage it (treatment representations). These representations lead them to identify and implement action plans to cope with the situation. Multiple sources contribute to shape illness and treatment beliefs: somatic sensations and deviations from normal daily functioning; prototypes and subjective representations of healthy functioning, illness and available treatments; opinions from experts and lay people, particularly family and friends. In the CSM, illness and treatment representations are articulated into five components: label and associated symptoms (identity); causes; course and duration (i.e. acute, chronic or cyclic timeline); expected effects on the individual's life (consequences); availability of interventions to affect the illness course through personal actions (controllability) and medical treatments (curability). In addition, individuals form beliefs on their affective response (emotional representations). Several studies highlighted that the representations of illness, treatment and emotions affect health outcomes directly, as well as indirectly through the adoption of specific coping strategies (i.e. efforts to manage health threats and consequent emotions) [37]. Most importantly, this self-regulation system is shaped by the individuals themselves in a subjectively meaningful and comprehensible way. Therefore, individual representations of and expectations about illness and treatment may or may not be congruent with clinicians' ones.

Based on these premises, the aim of the present study was to explore the literature investigating the care needs related to OD management of adult persons with OD and their informal caregivers during the last 20 years, using the SCF and the CSM as classification and interpretation framework respectively.

The following research question was formulated: What is known from the literature about the care needs related to OD management of adult persons with OD and their informal caregivers?.

We hypothesized that in the last 20 years the needs related to OD management of patients with OD and their informal caregivers (i) would cover physiological, behavioral, psychological and social aspects of patients' and caregivers' lives and (ii) would have been investigated mostly through qualitative research tools.

The decision to focus on the literature published in the last 20 years is related to the conceptual shift occurred at the beginning of the millennium in the understanding of disease and disability, leading researchers' and practitioners to pay more attention to patients' subjective perspective and illness experience. This shift clearly emerged in the International Classification of Functioning, Disability and Health (ICF), launched by the World Health Organization in 2001 [38] as a person-centered description of daily functioning, conceptualized as the interplay

between body structures and functions, environmental features and personal factors. In the same years, the central role of patients' illness experience was acknowledged in the Charter on Medical Professionalism [39] and in the SPIKES protocol, designed to support doctors in the stressful communication of bad news [40].

## Materials and methods

The present scoping review complies with the Preferred Reporting Items for Systematic Reviews and Meta-Analysis extension for Scoping Reviews (PRISMA-ScR) [41] (S1 PRISMA checklist). Following the methodological framework proposed by Arksey and O'Malley [42], the study was articulated into five stages: identifying the research question; identifying relevant studies; study selection; charting the data; collating, summarizing and reporting the results.

A "best fit" Framework synthesis approach was adopted [43,44]. This augmentative and deductive method involves the identification of an a priori relevant theoretical framework against which to map data from included studies. At the same time, the integration of an inductive perspective was deemed as relevant, in order to allow for shaping the chosen framework on participants' voices.

An operative protocol for conducting the review was agreed upon and shared by all team members, and registered on Open Science Framework (doi: 10.17605/OSF.IO/6ZCPK). The team was composed of seven members (3 speech and language pathologists, 1 phoniatrician, 2 psychologists, and 1 methodologist), contributing to the breadth and comprehensiveness of the review through their diversified expertise.

### Identifying the research question

The search strategy was based on the PICO (population, intervention, comparison, outcome) framework [45]. Adult persons with OD and/or their informal caregivers were identified as Population, needs assessment as Intervention, and unmet or satisfied OD-related needs as Outcome. The parameter Comparison was not deemed as applicable.

Appropriate thesaurus terms of the identified keywords and additional free terms were used to adapt the strings to each database. Search strategies are provided as additional file (S1 Table).

### Identifying relevant studies

On 16th January 2020, the International prospective register of systematic reviews (PROSPERO) [46] was checked for ongoing systematic reviews on the same topic, using the medical subject headings "Needs assessment" and "Health Services Needs and Demand". Results were screened by title and abstract to check if the population of interest included people with OD or their informal caregivers, yielding no relevant results. On 17th February 2021, the final search was conducted on the five electronic databases of PubMed, Embase, PsycINFO, Wiley Cochrane Library and Cinahl. Additional references were identified manually from the reference list of the publications eligible for full-text screening.

### Study selection

The identified records were imported into the software Rayyan.qcri [47] and duplicates were removed. Two reviewers independently screened the records for relevance, first based on title and abstract, and then on the full text. Disagreements were resolved through discussion and, if consensus could not be reached, by arbitration of a third rater. A track of the excluded full-text publications was kept in a table (available upon request from the corresponding author).

The eligibility of publications was based on ad-hoc inclusion and exclusion criteria regarding population and outcome of interest, language, type of study, year and type of publication. More specifically, inclusion criteria comprised (i) data collection from adult patients with OD and/or their informal caregivers; (ii) OD-related needs assessment; (iii) description of unmet and/or satisfied OD-related needs as reported by participants. Publications were excluded if (i) the full text was irretrievable (e.g. conference abstracts, or full text unavailability despite database and library search and contact with the corresponding author); (ii) data collection involved less than 10 participants or dyads (e.g. case studies); (iii) information to extract meaningful data was insufficient (e.g. inadequate description of the sample, context or study methodology); (iv) publication language was other than English or Italian; (v) hosting journals or books were not peer-reviewed; (vi) issue date was anterior to year 2000. Constraints regarding language and year of publication ensured the feasibility of full-text analysis and the inclusion of updated evidence, respectively. Studies including a reduced number of participants (<10), lacking meaningful information and not subjected to peer-review were excluded, in order to guarantee the highest possible methodological standards.

## Charting the data

The data-charting form was developed by the research team and independently piloted by two reviewers with the first six eligible full-text publications, in alphabetical order [48].

The following study characteristics were extracted: title; authors; year of publication; countries of data collection; participants' clinical characteristics; study design; tools for data collection and analysis. A critical appraisal of the eligible publications was performed through the Critical Appraisal Checklist for Qualitative Research tool developed by Joanna Briggs Institute (JBI) [49]. Critical appraisal was not considered an exclusion criterion, rather it informed results interpretation, highlighting potential flaws and need for further investigation. Participants' needs and the interventions aimed at their satisfaction were identified and extracted from the Results section only. For each extracted need, participants' exemplary verbatim quotations and the wording of questionnaire items referring to OD-related needs were reported for qualitative and quantitative studies, respectively, along with the authors' analytic interpretations.

## Collating, summarizing and reporting the results

A descriptive analysis of the study characteristics was performed. A convergent integrated approach was adopted in order to account for quantitative and qualitative data [50], as the review question could be answered with both types of evidence. Data synthesis occurred simultaneously, using qualitizing of quantitative data as process of data transformation. Integration of findings followed data transformation and relied upon a "best fit" Framework synthesis approach [43,44], suited for synthesizing results from qualitative studies within a given theoretical framework. For the present review, the SCF [27] was deemed as appropriate in order to categorize extracted OD-related needs. A preliminary need domain categorization was performed by two researchers and subsequently refined by adding lower classification levels, labelled as sub-categories. Emotional and psychological needs were included into the category "psychological needs", further articulated into sub-categories. When a quotation referred to multiple categories, a need for each one was counted. Exemplary quotations, as well as absolute (relative) frequencies of the needs and the studies in which they were cited were reported for each category and sub-category.

## Results

### Characteristics of the included studies

The search led to the identification of 2,881 records, to which 12 publications were added through manual search. After duplicate removal, 2,536 records were screened for eligibility by title and abstract, leading to the identification of 113 eligible publications. A careful reading of full texts led to the exclusion of 44 papers involving participants diagnosed with various pathologies, in which patients with OD could not be specifically identified. In addition, 34 papers were excluded because OD-related needs were not reported, 14 due to irretrievable full text, and 6 because of inadequate study design (e.g. case study) or insufficient information to extract meaningful data. Fifteen publications [21,50–63] were eventually included for data extraction. The study selection process is summarized in Fig 1 through a PRISMA flow diagram.

All the included studies were original works, 14 (93.3%) adopting a qualitative methodology and 1 (6.6%) quantitative methods. Data were collected through focus groups and interviews in the qualitative studies, and through a clinician-generated questionnaire in the quantitative study. Thirteen studies (86.6%) were published during the last 10 years, 7 of them (46.6%) during the last 5 years. All participants lived in Western countries, primarily English-speaking ones (11 studies, 73.3%). As usual in qualitative studies, sample sizes were rather small (participant n≤24), with one exception (n = 63) [58]. Most studies (10, 66.7%) were focused on patients' needs, whereas only 5 (33.3%) investigated the caregivers' perspective. The most frequent OD etiology (8 studies, 53.3%) was head and neck cancer; four studies (26.7%) involved participants with miscellaneous diagnoses and three other studies (20.0%) patients with neurodegenerative conditions. The characteristics of the studies included in the review are summarized in Table 1.

The critical appraisal with the JBI tools [49] proved satisfactory for all the included studies, with the only exception of Brockbank et al's [51] (S2 Table). In this study, the number of participants citing each need-related theme was not reported in the Results section, and conclusions were not clearly supported by the findings.

### Needs categorization

A total of 266 supportive care needs were extracted from the 15 publications. Overall, patients and caregivers primarily reported social (N = 77; 28.9%), practical (N = 67; 25.2%), informational (N = 55; 20.7%), and psychological (N = 54; 20.3%) needs. Psychological needs were cited in 12 (80.0%) different studies, while informational, practical and social ones emerged from 10 (66.7%) studies. Physical needs were reported only by patients (13 citations, 4.9%) across 7 studies (46.7%). Spiritual needs were not reported in the studies selected for the present review. Table 2 shows the distribution of need categories, separately for patients and caregivers. The former mostly focused on the psychological, social, practical and informational domains, the latter on practical and social needs.

Table 3 shows the distribution of care needs in categories and subcategories, with exemplary quotations for each subcategory. Results derived from each publication and an audit trail of the coding process are available as additional material (S3 and S5 Tables).

Within the physical domain, across clinical diagnoses patients expressed the need to consume tasty food and liquids [52,58,60] that could quench hunger and thirst, at the same time granting adequate nutritional status and swallowing safety [60,62,63]. They also expressed the need to attend SLP sessions, in order to learn techniques to improve their swallowing skills [55]. In the specific context of Motor Neuron Disease, participants relying on percutaneous

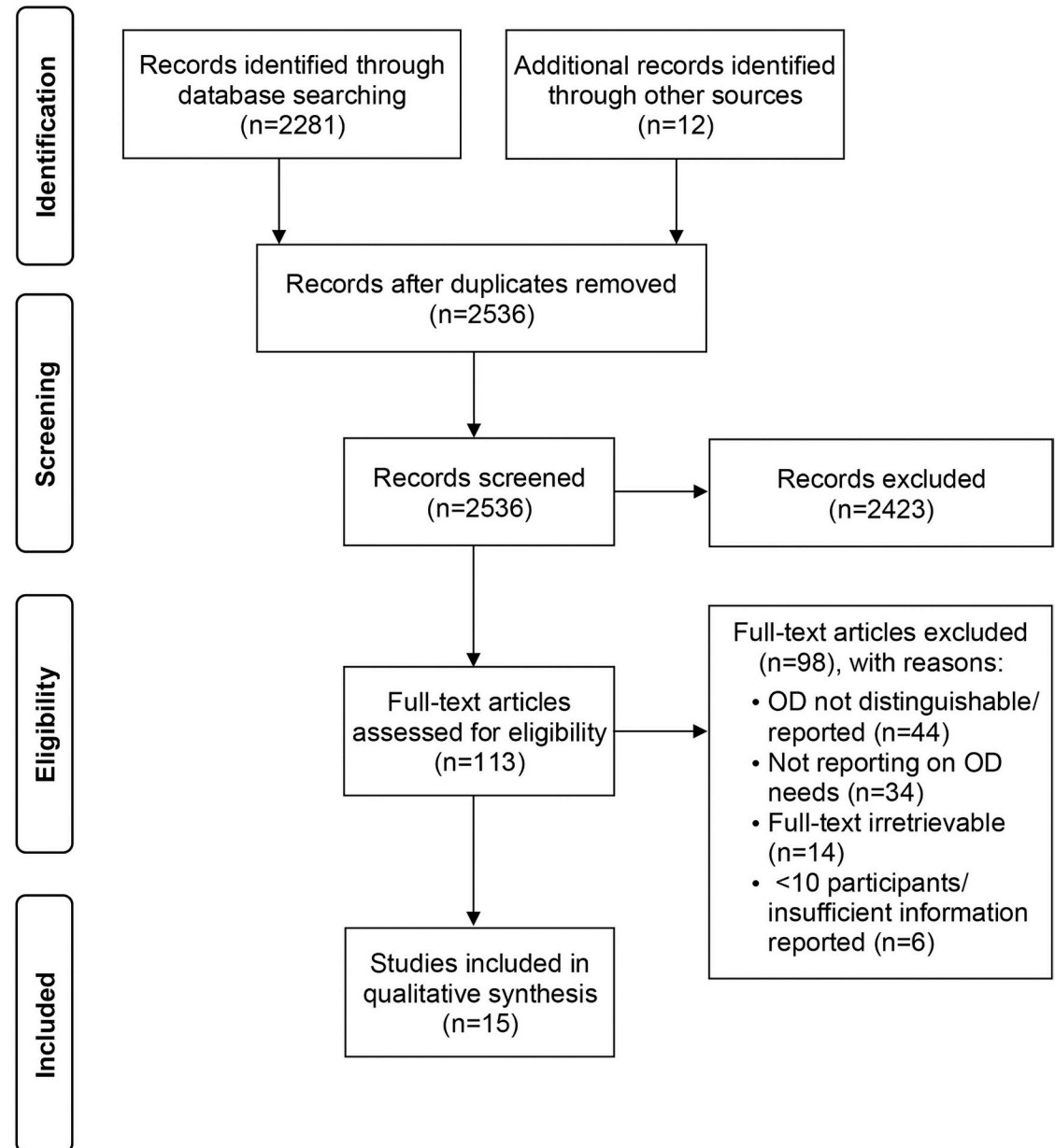

**Fig 1. PRISMA flow diagram summarizing the study selection process.** Adapted from Moher et al. [64].

endoscopic gastrostomy (PEG) as an alternative feeding method recalled their concern for body integrity preservation when they had been advised to undergo gastrostomy [57]. Patients with HNC, often experiencing mucositis as a consequence of radiotherapy, expressed the need to reduce pain while eating [52,63] and to adequately manage oral hygiene [63].

Patients and caregivers reported a wide range of practical needs, often formulated in terms of strategies and abilities allowing them to overcome difficulties. Two subcategories could be distinguished in this domain. Meal management included preparing food with adequate rheological and nutritional characteristics, assisting the patient in case of swallowing problems, investing additional time and thought on meals preparation and consumption, splitting lunch

**Table 1. Characteristics of the included studies.**

| Authors | Year | Title | Country | Participants | Patients' diagnosis | Study Design | Tools for data collection |
|---|---|---|---|---|---|---|---|
| Brockbank S, Miller N, Owen Sarah, Patterson JM | 2015 | Pretreatment information on dysphagia: exploring the views of head and neck cancer patients | UK | 24 Patients | HNC | Qualitative ecological | Focus group and interviews |
| Clarke G, Fistein E, Holland A, Tobin J, Barclay S, Barclay S | 2018 | Planning for an uncertain future in progressive neurological disease: a qualitative study of patient and family decision-making with a focus on eating and drinking | UK | 13 Patients 16 Caregivers | PD 3 (23.1%) MND 3 (23.1%) MS 2 (15.3%) PSP 3 (23.1%) Huntington's Disease 1 (7.7%) FTD 1 (7.7%) | Qualitative longitudinal | In-depth interview |
| Colodny N | 2005 | Dysphagic independent feeders' justifications for noncompliance with recommendations by a speech-language pathologist | USA | 63 Patients | Cerebrovascular accident 42 (66.7%) PD 5 (7.9%) COPD 4 (6.3%) Other diagnoses 12 (19.1%) | Qualitative ecological | In-depth interview |
| Govender R, Taylor SA, Smith CH, Gardner B | 2019 | Helping patients with head and neck cancer understand dysphagia: exploring the use of video-animation | UK | 13 Patients | HNC | Qualitative ecological | Focus group |
| Howard MM, Nissenson PM, Meeks L, Rosario ER | 2016 | Use of textured thin liquids in patients with dysphagia | USA | 20 Patients | Stroke 19 (95.0%) TBI 1 (5.0%) | Mixed cross-sectional | Clinician-generated questionnaire |
| Howells SR; Cornwell PL; Ward EC; Kuipers P | 2020 | Client perspectives on living with dysphagia in the community | Australia | 15 Patients | PD 6 (40.0%) Cerebrovascular accident 5 (33.3%) MSA 1 (6.7%) General ageing 1 (6.7%) HNC 1 (6.7%) Jaw fracture 1 (6.7%) | Qualitative ecological | Semi-structured interview |
| Howells SR; Cornwell PL; Ward EC; Kuipers P | 2020 | Living with Dysphagia in the Community: Caregivers "do whatever it takes." | Australia | 15 Caregivers | PD 7 (46.6%) Cerebrovascular accident 6 (40.0%) General ageing 1 (6.7%)HNC 1 (6.7%) | Qualitative ecological | Semi-structured interview |
| Larsson M, Hedelin B, Athlin E | 2007 | A supportive nursing care clinic: conceptions of patients with head and neck cancer | Sweden | 12 Patients | HNC | Qualitative longitudinal | In-depth interview |
| Lisiecka D, Kelly H, Jackson J | 2020 | 'This is your golden time. You enjoy it and you've plenty time for crying after': How dysphagia impacts family caregivers of people with amyotrophic lateral sclerosis–A qualitative study | Ireland | 10 Caregivers | MND | Qualitative ecological | Semi-structured interview |
| McQuestion M, Fitch M, Howell D | 2011 | The changed meaning of food: physical, social and emotional loss for patients having received radiation treatment for head and neck cancer | Canada | 17 Patients | HNC | Qualitative ecological | In-depth interview |
| Nund RL, Ward EC, Scarinci NA, Cartmill B, Kuipers P, Porceddu SV | 2014 | Carers' experiences of dysphagia in people treated for head and neck cancer: a qualitative study | Australia | 12 Caregivers | HNC | Qualitative ecological | Semi-structured interview |
| Nund RL, Ward EC, Scarinci NA, Cartmill B, Kuipers P, Porceddu SV | 2014 | Survivors' experiences of dysphagia-related services following head and neck cancer: implications for clinical practice | Australia | 24 Patients | HNC | Qualitative ecological | Semi-structured interview |
| Nund RL, Ward EC, Scarinci NA, Cartmill B, Kuipers P, Porceddu SV | 2014 | The lived experience of dysphagia following non- surgical treatment for head and neck cancer | Australia | 24 Patients | HNC | Qualitative ecological | Semi-structured interview |
| Ottosson S, Laurell G, Olsson C | 2013 | The experience of food, eating and meals following radiotherapy for head and neck cancer: a qualitative study | Sweden | 13 Patients | HNC | Qualitative ecological | In-depth interview |

*(Continued)*

**Table 1.** (Continued)

| Authors | Year | Title | Country | Participants | Patients' diagnosis | Study Design | Tools for data collection |
|---|---|---|---|---|---|---|---|
| Stavroulakis T, Baird WO, Baxter SK, Walsh T, Shaw PJ, McDermott CJ | 2014 | Factors influencing decision-making in relation to timing of gastrostomy insertion in patients with motor neuron disease | UK | 10 Patients 8 Caregivers | MND | Qualitative retrospective | Semi-structured interview |

HNC = Head and Neck Cancer; PD = Parkinson's Disease; MND = Motor Neuron Disease; MS = Multiple Sclerosis; PSP = Progressive Supranuclear Palsy; FTD = Frontotemporal Dementia; COPD = Chronic Obstructive Pulmonary Disease; TBI = Traumatic Brain Injury; MSA = Multiple System Atrophy.

and dinner into small but frequent meals, and giving practical assistance to patients during meals (e.g. feeding the patient or reminding him/her of behavioral strategies for meal consumption) [18,52–54,56,62,65]. Food availability referred to the restricted choice of safe and tasty food in healthcare facilities, at home, or during social occasions [18,53,54,61,63].

As concerns informational needs, three subcategories could be identified: the need for getting additional, clearer, and tailored information. Patients and caregivers wished to acquire better knowledge about symptom development, PEG advantages, health professionals' roles in OD management, and food recipes for OD [18,51–53,55–57,59,61,63]; they reported feeling overwhelmed by a large amount of information that was however unclear or difficult to understand [18,51,53,56,59]; they expected information to be more specifically tailored to their own experience [51,55,56,59,63].

Within Social needs, the most frequent subcategory was receiving support from caregivers or other family members, health professionals and other patients [18,51,55,56,61,63], followed by obtaining acceptance from family, friends, and community members [18,52,54,55,62,63,65], coping with changes in one's family and social roles [55,56,65], and experiencing concerns for the other member of the dyad [53–56,65].

Among the psychological needs, the predominant one was to cope with the emotions evoked by eating difficulties, such as fear and anger, and with a feeling of loss [18,51,54–57,61,63,65]. Furthermore, patients and caregivers reported problem-focused [18,51,56,57,59] and meaning-focused coping strategies, such as acceptance of the swallowing problem as a part of one's own identity, acceptance of a "new normal" condition, and recognition of changes in one's own social roles [18,54,56,63]. In addition, patients and caregivers expressed the need to better connect the information received about symptom development with their own daily experience of OD [18,56,63]. Some patients reported the need to be motivated at eating, due to loss of eating desire [55,62,65], while others hoped to be able to enjoy food again [52,55,63]. Similarly, caregivers expressed the importance of maintaining a positive attitude in order to

**Table 2. Patients' and caregivers' needs categorization.**

| Needs Category | Patients with OD | | Caregivers | |
|---|---|---|---|---|
| | Needs (N = 163) N (%) | Studies (N = 15) N (%) | Needs (N = 103) N (%) | Studies (N = 15) N (%) |
| Informational | 33 (20.2%) | 7 (46.7%) | 22 (21.4%) | 4 (26.7%) |
| Psychological | 35 (21.5%) | 9 (60.0%) | 19 (18.4%) | 3 (20.0%) |
| Social | 48 (29.4%) | 6 (40.0%) | 29 (28.2%) | 5 (33.3%) |
| Practical | 34 (20.9%) | 6 (40.0%) | 33 (32.0%) | 4 (26.7%) |
| Physical | 13 (8.0%) | 7 (46.7%) | – | – |

Absolute (relative) frequencies of the need categories and the studies in which they were cited by patients with OD and their caregivers.

**Table 3. OD-related needs classification with subcategories and exemplary citations.**

| Category | Subcategory | n (%) | ref (%) | Description | Citations |
|---|---|---|---|---|---|
| **Physical needs** | | 7 (46.7%) | 13 (4.9%) | | |
| | Taste | 3 (20.0%) | 3 (1.1%) | Food does not taste good | "Don't give me the thick liquids.. . . I don't like the way they taste, and I just don't like them. I can't be bothered." [58] |
| | Swallowing skills | 1 (6.7%) | 2 (0.8%) | Learning techniques to improve swallowing skills | "I keep clearing my throat of saliva and I have to be content with swallowing more slowly. I just swallow more deliberately and I have to be careful I don't choke myself. . . Don't swallow too fast or too much at a time." [55] |
| | Food function | 3 (20.0%) | 4 (1.5%) | Food and liquids do not quench hunger and thirst, fail to maintain adequate nutritional status or are not safe to be administered | "You kind of live on [supplements] a bit, that sort of keeps you going." [63] |
| | Body integrity | 1 (6.7%) | 1 (0.4%) | Maintaining body integrity, do not want to be messed about | "Yeah, s/he didn't want nothing like that [PEG], no, s/he didn't want to be messed about." [57] |
| | Oral hygiene | 1 (6.7%) | 1 (0.4%) | Maintaining oral hygiene | "I have to. . .of a morning. . .and regularly throughout the day. . .have a rinse of bicarb and. . .salt together." [63] |
| | Pain | 2 (13.3%) | 2 (0.8%) | Reducing pain/burning in the mouth or at swallowing | "It was painful swallowing I had to. . . take some. . .medication. . .so I can eat." [63] |
| **Practical needs** | | 10 (66.7%) | 67 (25.2%) | | |
| | Food availability | 8 (53.3%) | 29 (10.9%) | Restricted choice of food with adequate characteristics (safe to swallow and taste) | "And of course the eating habits. It's quite hard, because we used to, my husband and I, used to go out every Wednesday with friends for dinner. I just started to go back. And of course I'll have my little soup, and it depends what kind of soup they have, sometimes I can't even eat it and will just have a cup of tea or a cup of coffee and I drink my water." [62] |
| | Meal management | 7 (46.7%) | 38 (14.3%) | Managing to prepare food with adequate rheological and nutritional characteristics | "Most things . . . have . . . to be very soft." [63] |
| | | | | | "I buy cream and double cream on a very regular basis [. . .] for calories [. . .]. I might make the soup but I'd put a lot of cream in [. . .] to keep the calories because [. . .] the first thing, one of the Motor Neuron nurses she says just keep the weight on [. . .] I usually buy the super whole milk [. . .]. When I go shopping I think of calories most of the time." [54] |
| | | | | Additional time/thought necessary for meal preparation and consumption | "Mealtimes. . .a lot more thought went into it." [18] |
| | | | | Assisting the patient in case of choking/swallowing problems at meals | "How do I cope with this, what do I do. . .to make it so he doesn't choke on his food?" [18] |
| | | | | Splitting lunch/dinner into frequent small meals | "I have so many little meals a day and. . .snacks." [63] |
| | | | | Practical assistance during meals (feeding the patient/reminding the patient behavioral strategies for meal consumption) | "There's all those sorts of strategies that we need to be reminding her of. She needs to take single sips. . . she needs to take it a little bit slower and she tends to chug it very fast. We have in the past done chin tuck, she doesn't remember to do that all the time. . . Getting her to have a fluid flush, like have a drink with her meals." [56] |
| **Informational needs** | | 10 (66.7%) | 55 (20.7%) | | |
| | More information | 10 (66.7%) | 29 (10.9%) | Received insufficient information about symptoms development/PEG advantages/ health professionals' role/food recipes for OD | "Well I think I would have probably gone for the PEG earlier, if I'd have known. Obviously when the PEG was going in I weren't that sure because I didn't really know about it then, but actually having it done has made my job a lot easier. I mean, it was difficult feeding him/her [the patient] before that, you know." [57] |

*(Continued)*

**Table 3.** (Continued)

| Category | Subcategory | n (%) | ref (%) | Description | Citations |
|---|---|---|---|---|---|
| | | | | | "I think you do need to know about long term because conventionally you think treatment, oh I'm better now I've had treatment I must be better but actually no I'm not that's for me the beginning of the horrors really and the treatment I thought yes great then two days later I really crashed and ended up with this (nasogastric tube)." [51] |
| | Clearer information | 5 (33.3%) | 15 (5.6%) | Information received is unclear | "I just got a photocopy of the diagram thing [referring to how information about swallowing was provided during his own pre-treatment counseling]. It's much clearer, what's going on, when you can see it from a proper video of it." [59] |
| | | | | Received too much information | "[Health professionals] think their knowledge is everybody's knowledge and it's not. They [have] got to use...patient language [...] The explanations before treatment...sore throat, you may not be able to produce saliva...was put in a medical way but the reality is the human element. No one tells you about that." [18] |
| | Tailored information | 5 (33.3%) | 11 (4.1%) | Information received is not tailored | "They should give you the option you know straight away would you like to know straightaway here and now or would you like a few days to go home and think about it and let it sink in and then come back and you can be told." [51] |
| **Social needs** | | 10 (66.7%) | 77 (28.9%) | | |
| | Social acceptance | 6 (40.0%) | 15 (5.6%) | Acceptance from family members/friends/ others | "Some of your friends, if you've got a restricted diet, they won't invite you to a meal because they... don't know what you're going to be able to eat." [63] |
| | Change in social roles | 3 (20.0%) | 11 (4.1%) | Changes in one's own family/social image | "I used to like going out...I don't do that anymore. So far as my social life goes, I don't have any" [65] |
| | Social support | 6 (40.0%) | 33 (12.4%) | Social support from other patients | "I would have liked to have talked to someone who has been through the same thing, you can't replace that." [63] |
| | | | | Social support from health professionals | "There's ups and downs all the time there might be a week when you almost feel nothing and suddenly you are back on square one. On these occasions the contact with the nurse is invaluable. To have somebody to ask if this is normal, that this is how it's supposed to be, eases your mind." [61] |
| | | | | Social support from caregivers/other family members | "I've needed someone to understand about this [eating difficulties"; "[carer's] been really understanding on the [eating] side of things, she says 'take your time"; "the family's not in your mouth, they don't know what's going on." [63] |
| | Dyad member's well-being | 5 (33.3%) | 18 (6.8%) | Concern regarding the needs of the other member of the dyad | "This is a we thing not a me thing because it affects [carer]... it affects [carer] as much as it affects me, ...how I am." [65] |
| **Psychological needs** | | 12 (80.0%) | 54 (20.3%) | | |
| | Emotion-focused coping | 9 (60.0%) | 22 (8.3%) | Coping with emotions and feelings of loss | "And, and it was really just fear of not knowing, wasn't it, and fear of having it done. It wasn't being frightened of having it done, it was just so big a change to your life, isn't it?" [53] |
| | | | | | "It has meant a lot to her [wife] and of course indirectly to me as well. Because there are strange reactions as it's difficult to explain. It's like a fear on her side, a fear that sometimes passes over to anger, which can't be explained. Sometimes she could be angry with me for being illy after talking to the nurse everything went very, very well." [61] |
| | Motivation | 3 (20.0%) | 6 (2.3%) | Lack of motivation/desire to eat | "It got to a point where you just lost interest, you lost your appetite, you lost your desire to eat" [65] |
| | Awareness | 3 (20.0%) | 3 (1.1%) | Becoming aware of symptoms development/changes in daily life | "I didn't realize how bad the eating was going to get." [18] |

(*Continued*)

**Table 3.** (Continued)

| Category | Subcategory | n (%) | ref (%) | Description | Citations |
|---|---|---|---|---|---|
| | Problem-focused coping | 5 (33.3%) | 7 (2.6%) | Problem-focused coping strategies | "When I'm very tired I have to be more careful about my swallowing, and make sure that I chew my food thoroughly, but at the moment it doesn't really affect me greatly . . . I do have episodes of choking when I'm drinking but other than that, no, it's not a problem" [57] |
| | Acceptance | 5 (33.3%) | 10 (3.8%) | Acceptance of the "new normal" | "You have to live with the new life. . .and the new life is that meal preparation is different from anybody else's or different from what it used to be." [18] |
| | | | | Considering the swallowing problem as little | "You just learn to live with it, you don't even think about it now it's just a part of your diet." [63] |
| | | | | Acceptance of changes in social roles | "When you. . .[look] after somebody over a length of time you do become the carer and you do become the parent." [18] |
| | Personal Growth | 2 (13.3%) | 1 (0.4%) | Caring experience as an occasion of personal growth | "It's good to care for someone, it definitely changes you. For the better." [56] |
| | Positive attitude | 2 (13.3%) | 2 (0.8%) | Maintaining a positive attitude | "You have to be positive and find foods that they can cope with and just go with that." [18] |
| | Hope | 3 (20.0%) | 3 (1.1%) | Cultivating hope | "I wish it would not burn in my mouth when I eat, that would be amazing. But I do not know if it will ever go away, it may take time." [52] |

Categories and subcategories of OD-related needs with absolute (relative) frequencies, sub-category descriptions, and examples of citations.

n = number of studies in which the category/subcategory were cited.

ref = number of needs retrieved for each category/subcategory.

overcome eating-related difficulties [18,54] and to consider the caregiver role as an opportunity for personal growth [56].

## Interventions to address care needs

Only 3 studies [59–61] described interventions to address the needs of patients, whereas none was focused on caregivers. The usefulness of textured thin liquids in satisfying the physical needs for maintaining hydration and quenching thirst was investigated among persons with stroke and traumatic brain injury [60]. An educational intervention based on video-animation was developed to help patients with HNC understand normal and abnormal swallowing mechanisms [59]. Findings highlighted the usefulness and acceptability of this intervention for enhancing patients' understanding of swallowing pathophysiology, as well as their active engagement in prophylactic interventions after treatment. The perceived significance of a supporting nursing care clinic for patients with HNC was longitudinally assessed before, during and after radiotherapy [61]; variations were detected according to patients' positioning along the care trajectory, and nature and severity of the problems experienced. Authors claimed that this intervention met patients' needs of knowledge, care, support, and adaptive coping with emotions.

## Discussion

Overall, the present scoping review shed light on the dearth of research studies focused on the care needs of persons with OD and their informal caregivers.

A positive trend was nevertheless observed over time, as most of the retrieved studies were published in the last decade. As expected, data were primarily collected through a qualitative methodology, that offers the opportunity to delve into complex and underexplored phenomena such as unmet needs. On the other hand, the small sample size and the non-standardized

tools for data collection, typical of qualitative studies, entail limitations in results generalizability and reproducibility [66]. These limitations must be taken into account when interpreting the findings emerged from the present review. The development and validation of a standardized tool to specifically assess the needs of patients with OD and their caregivers could contribute to enhance research rigor in this domain.

All the included publications demonstrated a satisfactory methodological quality, with the exception of one study, in which authors' conclusions were only partially supported by participants' voices. Only participants' verbatim quotations reported in the Results section were however considered as units of analysis; therefore, we argue that the unsatisfactory critical appraisal emerging in this specific case would not impact the results of the review. In fact, drawing conclusions about the effectiveness of the interventions retrieved goes beyond the scope of the present review.

All the examined studies were conducted in Western countries. This finding may be related to one of the inclusion criteria, namely the use of English or Italian as publication language. It should be however noted that studies from Eastern Asian countries retrieved through the screening process had to be excluded due to their focus on OD treatment (e.g. acupuncture) rather than related needs [67,68]. It may be hypothesized that in these cultures patients' and caregivers' needs represent a rather private issue, not liable to investigation within the medical literature.

Over half of the included studies involved participants with a diagnosis of head and neck cancer. Such an over-representation may be related to the high relevance of swallowing disorders and related implications for these patients. OD may represent a less relevant concern for patients with neurodegenerative diseases and their caregivers, compared to the generalized motor problems, weakness and fatigue, and autonomy loss they have to cope with [55,56,69,70]. Finally, participants experiencing acute neurologic pathologies may perceive OD as a temporary symptom, requiring immediate clinical management rather than implying long consequences. Further studies are needed to more correctly evaluate the importance of OD, net of other symptoms, for patients and caregivers in different clinical conditions.

Despite evidence of increased physical and emotional burden levels among informal caregivers of elderly patients with OD [20], only few of the included studies investigated informal caregivers' needs, a topic deserving further exploration.

Overall, the pursuit of the research aim–identifying perceived care needs–led to the retrieval of a very limited number of studies, reflecting a substantial lack of attention to patients' and caregivers' perspectives in both research and clinical domains. Nevertheless, as expected, a broad variety of OD-related needs and strategies for their satisfaction were identified by both patients and caregivers. Informational, social, and psychological needs were equally cited by patients and caregivers, whereas physical needs were reported only by patients and practical needs concerning strategies to manage OD were primarily quoted by caregivers.

Traditionally, OD research has been prominently focused on swallowing physiology, OD clinical assessment and management field [71]. Since the introduction of the ICF, more attention was paid to the manifold implications of OD [9–11], through scales investigating the psychological and social dimensions, patients' perception of OD physical symptoms, and HRQOL [72,73]. Despite the usefulness of these quantitative measures towards a multidimensional standardized OD assessment, they do not allow for evaluating the importance attributed by individual patients to each OD related impairment or problem. Considering that need perception arises from the gap between the perceived severity of functional loss and the value attributed to the lost function [26], qualitative research instruments such as interviews and focus groups appear as the most appropriate instruments to investigate OD-related needs; related

findings may subsequently guide the development of questionnaires specifically targeting those needs.

To the best of our knowledge, this is the first scoping review aimed at summarizing the available evidence on patients' and caregivers' OD-related needs, from a person-centered perspective. At the conceptual and interpretive levels, the lack of a solid, unifying and person-centered theoretical model guiding research hypotheses and result interpretation emerged from the analysis of the reviewed studies. In the present review, the Supportive Care Framework (SCF) [27], previously used for classifying needs reported by patients diagnosed with stroke and ALS [33,34], proved to be a satisfactory theoretical framework to categorize OD-related needs. In order to fit the results of the present review, only slight changes were brought to the framework. In particular, emotional and psychological needs were merged into the category "psychological needs" as, from a disciplinary perspective, emotions belong to psychological domain, together with cognition and motivational processes. In addition, in order to better inform future research and clinical practice, the SCF categories were articulated into OD specific sub-categories.

Based on the review findings, we also suggest the potential of the Common-Sense Model of Self-Regulation (CSM) [35,36] as a suitable conceptual and interpretive framework to guide research on patients' and caregivers' needs. The broad range of care needs emerged from the present review can be fruitfully interpreted through the CSM. Informational needs can be related to illness or treatment representations. For example, patients' and caregivers' difficulties in building a mental representation of OD based on their own experience, in terms of illness label, associated symptoms and timeline may elicit the need for more detailed, clearer or tailored information about these issues. Similarly, in order to form an adequate mental representation of treatment consequences, patients and caregivers may need more detailed information about PEG advantages. Furthermore, informational needs may guide action, as highlighted by the patients' and caregivers' need for more detailed and tailored information on OD-adapted recipes and OD management during holidays.

Practical needs may instead emerge from limitations in individual and environmental resources, such as restricted availability of safe and tasty food, or lack of competences in preparing meals.

Physical needs, such as maintaining body integrity or reducing pain while eating, may be linked to treatment representation (PEG and radiotherapy, respectively) and related consequences. The need for nutritionally adequate meals or proper oral hygiene may relate to the representation of OD causes (undernutrition could worsen OD through decreased muscles strength) and controllability (undernutrition could accelerate OD worsening in chronic conditions and slow recovery in acute ones).

Social needs may be primarily related to the representation of OD consequences. For instance, the changed meaning of food [12] may give rise to the need for social support and for changes in family and social roles. The physical and emotional caregivers' burden consequent to OD [18,20] may lead patients to worry about the wellbeing of their caregiver.

Finally, psychological needs may be related to different components of the CSM. The need to cope with negative emotions and feelings of loss—frequently reported by persons with OD [9]—may be associated with OD emotional representation. Problem-focused strategies acted by OD patients may be linked to the CSM action plans aimed at best controlling OD symptoms and consequences. The need to accept a "new normal", to maintain a positive attitude and to cultivate hope may be related to the representation of illness coherence.

Spiritual needs (i.e. "needs related to the meaning and purpose in life to practice religious beliefs" [27]), though not retrieved in the present review, could also be included in the CSM component of illness coherence. Studies involving a wider range of countries and cultural

groups could contribute to a more comprehensive understanding of the potential role of spiritual needs in OD patients and informal caregivers.

Besides shedding light on patients' and caregivers' illness and treatment representations, the CSM could be a suitable framework at the intervention level. By taking into account the reciprocal interplay between mental representations and action plans, the former contributing to shape the latter and vice-versa, researchers and clinicians could design needs-driven interventions [59] based on the global representation of illness and treatment subjectively built by patients and caregivers.

## Limitations and future directions

The present review is not exempt from limitations. Although it was aimed at summarizing the available evidence concerning the care needs of patients with OD and their informal caregivers, the eligible studies were limited in number and prominently involving persons with HNC. The results might thus be partial and non-generalizable. In addition, few of the retrieved studies explored caregivers' needs, a topic requiring more thorough investigation. A caveat should be also expressed when interpreting patients and caregivers' OD-related needs. The OD etiology–widely varying in the examined studies–might represent a confounding factor, leading participants to quote needs only partially referable to OD symptoms and consequences. In the attempt to overcome this problem, only needs strictly related to swallowing and feeding were extracted in the present review. In addition, in order for results to be useful for clinical practice, besides unmet needs per se we also extracted and classified strategies envisaged or acted by patients and caregivers to satisfy them.

As concerns future directions, we have proposed an interpretation of the review findings based on the CSM, suggesting its potential as a framework to understand care needs from the patients' and caregivers' subjective perspectives. The inclusion of this model in future studies, both as a conceptual and interpretive framework, and as an approach to data collection through the related scales (e.g. the Revised Illness Perception Questionnaire, IPQ-R [74]), could elucidate the relationship between illness beliefs, need perception and behavioral strategies enactment among patients and caregivers, and across clinical conditions.

More specific tools for OD needs assessment should also be developed and validated, in order to foster rigorous and comparable research on larger groups of patients and informal caregivers, differing in terms of diagnosis and cultural contexts. These advancements could contribute to designing authentically patient-centered interventions, that could be compared to clinician-driven interventions in terms of efficacy and client satisfaction.

## Supporting information

**S1 PRISMA checklist. PRISMA-ScR checklist.**
(DOCX)

**S1 Table. Search strategy.**
(DOCX)

**S2 Table. Critical appraisal.**
(XLSX)

**S3 Table. Results of individual sources of evidence.**
(XLSX)

**S4 Table. Funding of included sources of evidence.**
(XLSX)

**S5 Table. Audit trail of OD-related needs categorization.**
(XLSX)

## Acknowledgments

The authors would like to acknowledge the speech and language pathologist, Dr Thais Alves, for the valuable contribution in the initial development of the search strategies and the charting data form.

## Author Contributions

**Conceptualization:** Aurora Ninfa, Antonio Schindler, Antonella Delle Fave.

**Data curation:** Aurora Ninfa.

**Formal analysis:** Aurora Ninfa, Valeria Crispiatico, Antonella Delle Fave.

**Funding acquisition:** Marta Bassi, Giovanni Casazza, Antonio Schindler.

**Investigation:** Aurora Ninfa, Valeria Crispiatico, Antonella Delle Fave.

**Methodology:** Aurora Ninfa, Giovanni Casazza, Antonio Schindler, Antonella Delle Fave.

**Project administration:** Aurora Ninfa.

**Software:** Aurora Ninfa.

**Supervision:** Antonio Schindler, Antonella Delle Fave.

**Visualization:** Aurora Ninfa, Antonio Schindler, Antonella Delle Fave.

**Writing – original draft:** Aurora Ninfa.

**Writing – review & editing:** Aurora Ninfa, Valeria Crispiatico, Nicole Pizzorni, Marta Bassi, Giovanni Casazza, Antonio Schindler, Antonella Delle Fave.

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
