## [Decision Letter · Decision Letter 0]

21 Apr 2021

PONE-D-21-06132

The care needs of persons with oropharyngeal dysphagia and their informal caregivers: a scoping review

PLOS ONE

Dear Dr. Ninfa,

Thank you for submitting your manuscript to PLOS ONE. After careful consideration, we feel that it has merit but does not fully meet PLOS ONE’s publication criteria as it currently stands. Therefore, we invite you to submit a revised version of the manuscript that addresses the points raised during the review process.

See comments below.

We look forward to receiving your revised manuscript.

Kind regards,

Andrew Soundy

Academic Editor

PLOS ONE

Journal Requirements:

In the Methods section, please clearly indicate which checklists were used from the  Joanna Briggs institute was used for the quality assessment of the included articles

We note that this manuscript is a systematic review or meta-analysis; our author guidelines therefore require that you use PRISMA guidance to help improve reporting quality of this type of study. Please upload copies of the completed PRISMA checklist as Supporting Information with a file name “PRISMA checklist”.

We noted in your submission details that a portion of your manuscript may have been presented or published elsewhere. The authors acknowledge that preliminary findings of this work were presented as a poster at the European Society for Swallowing Disorders (ESSD) 10th virtual congress (8th-10th October 2020). The poster will be published as conference proceedings in Dysphagia. We provide the ESSD abstract as attachment.

Please clarify whether this poster was peer-reviewed and formally published. If this work was previously peer-reviewed and published, in the cover letter please provide the reason that this work does not constitute dual publication and should be included in the current manuscript.

Additional Editor Comments :

General

I think there is a lot of aims and I believe careful consideration is needed to what you want to achieve from this review. I have listed out considerations per section please consider and justify answers in your response letter. Failure to do this could result in rejection at the next stage.

Introduction

Where you state one of the few attempts to classify patients needs line 69 – Please contextualise for the reader e.g., if a reader searched for frameworks on google the following come up – so why selected this one for your OD review?

https://www.liebertpub.com/doi/full/10.1089/jpm.2020.0435

https://journals.sagepub.com/doi/abs/10.1177/073346489501400104

https://europepmc.org/article/med/18326380

http://canadianoncologynursingjournal.com/index.php/conj/article/view/248

https://cjasn.asnjournals.org/content/14/4/635?WT_MC_ID=TMDPC&utm_campaign=TMDPC&utm_medium=cpc&utm_source=TrendMD

https://apps.who.int/iris/bitstream/handle/10665/274628/9789241514033-eng.pdf

https://bmjopen.bmj.com/content/10/7/e034970.abstract

again context for the need for your review in terms of what is known, not known and further needs to be understood around this area e.g., if you searched for reviews in the area this comes up on google – so the reader needs to understand the context of your review

https://link.springer.com/content/pdf/10.1007/s00134-020-06126-y.pdf

https://link.springer.com/article/10.1007/s12603-020-1377-5

https://link.springer.com/article/10.1007/s00455-017-9844-9

https://journals.sagepub.com/doi/full/10.1177/0145561320917795

https://link.springer.com/article/10.1007/s00134-020-06189-x

carers

https://pubs.asha.org/doi/abs/10.1044/2020_PERSP-20-00067

you need to be clear why you have focused on the common sense illness model – it comes out of the blue

There are lots of aims from line 77, then research questions from line 82 then hypothesis from line 85 – please give one general aim or purpose as a focus at the end of the introduction giving a greater amount of space to the rationale needed above – else I think it is confusing for the reader

Methods

Was there a protocol published for this review e.g., PROSPERO?

I am not sure about the definitions placed here consider the introduction or a supplementary file. Can you use PICOS, SPIDER or another acronym to identify eligibility criteria first? Then this would give guidance for your key words also

For the searching please just state when the final search took place and ignore the fact you searched earlier – please update the abstract also

I found the eligibility criteria hard to follow – you identify constraints line 142-143 and then more specific criteria - can you just have one place where you identify criteria and split criteria into inclusion and exclusion – please do not have the opposite criteria for exclusion just other criteria

Give rationale for your criteria e.g., why after 2000? Why did studies have to involve less than 10 participants?

As you are dealing with a lot of qualitative studies can you provide a methodology and paradigmatic position upfront for your review

The reader will need to understand how you came up with your results and an audit trail is needed in a supplementary file to show stages of analysis

The analysis will have to address integration between qualitative and quantitative data – as you have different designs of studies

How was critical appraisal used for different types of study and what impact does it have on the results. Please really consider this

Results

Can you add some sub-headings for needs categorisation

The reader needs to understand how study quality influenced results – especially for interventions

You need to consider the link to the common sense model of illness

Discussion

The discussion lacks references can you check list and develop it further

The common sense illness model if defined here which is too late – does your data really show this model from anticipating threat to coping ? for both patients and carers? I think this distracts the reader quite a lot from what was coming through the data - consider and justify in your response if, and how much time is allocated to this

Reviewers' comments:

Reviewer's Responses to Questions

**Comments to the Author**

1. Is the manuscript technically sound, and do the data support the conclusions?

Reviewer #1: Partly

2. Has the statistical analysis been performed appropriately and rigorously? 

Reviewer #1: Yes

3. Have the authors made all data underlying the findings in their manuscript fully available?

Reviewer #1: Yes

4. Is the manuscript presented in an intelligible fashion and written in standard English?

Reviewer #1: Yes

5. Review Comments to the Author

Reviewer #1: Thank you for the opportunity to review this manuscript. This is an important area of research that has received little attention to date. The manuscript is well written and the methods are clearly described. My main concern regarding the manuscript is that there appears to be a number of relevant studies that have not been included. Perhaps they have been excluded for valid reasons but this is not clear based on the inclusion criteria. Publications such as Patterson et al., 2013; Howells et al., 2020a and Howells et al., 2020b. If these studies have been excluded, it is suggested that the authors carefully consider the wording of their inclusion/exclusion criteria to further highlight why these types of studies have been excluded. This is to ensure that the study methodology is reproducible. If there are studies that have been missed, such as the above, then the results will need to be reanalysed and the discussion may need to be reworked depending on whether changes to the results are required.

In addition, it was noted that the quote provided in Table 3 - Information needs (page 17) reference 18 is incorrect as this quote is not from this article. It is suggested that the authors double check all included quotes and the respective studies.

6. PLOS authors have the option to publish the peer review history of their article (what does this mean?). If published, this will include your full peer review and any attached files.

Reviewer #1: No

---

## [Author Response · Author response to Decision Letter 0]

18 Jun 2021

Thank you very much for your review of our manuscript and for the valuable comments and suggestions. We have provided our responses to your comments in this letter, and revised the manuscript accordingly. 

Journal Requirements:

In order to comply with PLOS One style requirements, on line 4 the subtitle first letter has been capitalized. In the abstract, bold was removed from paragraph names and they now appear formatted as plain text (lines 16, 20, 27, 33, 43 and 48). In addition, Figure 1 was cited as “Fig 1” both in the text and in the figure caption (lines 268 and 275). 

2. In the Methods section, please clearly indicate which checklists were used from the Joanna Briggs institute was used for the quality assessment of the included articles

On lines 232 and 233, we specified that the Critical Appraisal Checklist for Qualitative Research tool by Joanna Briggs Institute was used for critical appraisal of the included studies. The name of the checklist is also reported within the supporting information file “S3 Table”.

3. We note that this manuscript is a systematic review or meta-analysis; our author guidelines therefore require that you use PRISMA guidance to help improve reporting quality of this type of study. Please upload copies of the completed PRISMA checklist as Supporting Information with a file name “PRISMA checklist”.

In fact, we undertook a scoping review, since the breadth of our research aim, the number of studies and the typology of data collected on the target topic did not allow us to undertake a systematic review. During the submission of the manuscript’s original version, we had uploaded a copy of the completed PRISMA checklist for Scoping Reviews (PRISMA ScR) as supporting information named “S1 Table”. Based on the Reviewer’s suggestion, we have renamed it as “S1 PRISMA checklist”. The reference in the manuscript was changed accordingly (lines 149 and 783).

4. We noted in your submission details that a portion of your manuscript may have been presented or published elsewhere. “The authors acknowledge that preliminary findings of this work were presented as a poster at the European Society for Swallowing Disorders (ESSD) 10th virtual congress (8th-10th October 2020). The poster will be published as conference proceedings in Dysphagia. We provide the ESSD abstract as attachment”.

Please clarify whether this poster was peer-reviewed and formally published. If this work was previously peer-reviewed and published, in the cover letter please provide the reason that this work does not constitute dual publication and should be included in the current manuscript.

The abstract of the poster presented at the European Society for Swallowing Disorders (ESSD) 10th virtual congress was reviewed by 3 members of the scientific committee and has not yet been published.

In our opinion the poster does not constitute dual publication, since it is a conference presentation that will be published in the conference proceedings, as allowed by PLOS ONE policy. In addition, compared to the present manuscript, only preliminary results derived from a partial list of the documents used for the present work were presented. Moreover, the poster format did not allow us to provide details about methodology, which constitutes the essence of a literature review. Moreover, only the broader categories of needs were reported, and results were not interpreted through a theoretical framework, such as the Common-Sense Model of Self-Regulation. Finally, the PRISMA flow diagram presented at the ESSD congress differs from the figure included in the present manuscript, since the final search leading to the inclusion of more recently published articles was conducted just before submitting to PLOS ONE. 

Additional Editor Comments:

General

I think there is a lot of aims and I believe careful consideration is needed to what you want to achieve from this review. I have listed out considerations per section please consider and justify answers in your response letter. Failure to do this could result in rejection at the next stage.

We thank the Editor for the thorough revision of our manuscript and the useful suggestions. We provide point-by-point answers below each comment.

Introduction

Where you state one of the few attempts to classify patients needs line 69 – Please contextualise for the reader e.g., if a reader searched for frameworks on google the following come up – so why selected this one for your OD review?

https://www.liebertpub.com/doi/full/10.1089/jpm.2020.0435

https://journals.sagepub.com/doi/abs/10.1177/073346489501400104

https://europepmc.org/article/med/18326380

http://canadianoncologynursingjournal.com/index.php/conj/article/view/248

https://cjasn.asnjournals.org/content/14/4/635?WT_MC_ID=TMDPC&utm_campaign=TMDPC&utm_medium=cpc&utm_source=TrendMD

https://apps.who.int/iris/bitstream/handle/10665/274628/9789241514033-eng.pdf

https://bmjopen.bmj.com/content/10/7/e034970.abstract

We thank the Editor for raising this point and providing all the references. In order to clarify the context of our review to the reader, we broadened the introduction presenting the relevant references on existing needs frameworks (WHO, 2018; Nelson et al., 2021; Lam et al., 2019; Diwan & Moriarty, 1995; Hollander & Prince, 2008). We explained the reasons for choosing the Supportive Care Framework (Fitch, 2008), as the only one adopting an in-depth person-centered approach to the specific content of patients’ needs. All the other frameworks are organization-centered, rather than person-centered, and they are focused on barriers and facilitators for health services utilization or on the promotion of a better healthcare organization (lines 84-94). We did not mention the work by Youssef et al. (2020) because it explores patients’ experiences of integrated care, which is beyond the scope of the present review.

again context for the need for your review in terms of what is known, not known and further needs to be understood around this area e.g., if you searched for reviews in the area this comes up on google – so the reader needs to understand the context of your review

https://link.springer.com/content/pdf/10.1007/s00134-020-06126-y.pdf

https://link.springer.com/article/10.1007/s12603-020-1377-5

https://link.springer.com/article/10.1007/s00455-017-9844-9

https://journals.sagepub.com/doi/full/10.1177/0145561320917795

https://link.springer.com/article/10.1007/s00134-020-06189-x

carers

https://pubs.asha.org/doi/abs/10.1044/2020_PERSP-20-00067

Mirroring the considerations reported above, we expanded the introduction with the most relevant literature reviews in the OD field. As suggested by the Editor, we included the works by Duncan et al. (2020), Zuercher et al. (2020), Ballesteros-Pomar et al. (2020), Jones et al. (2018), Shune & Namasivayam-MacDonald (2020). We also added the reference to the review on the production of dysphagia food using texture modification technologies (Sungsinchai et al., 2019) and on the impact of modified-texture diets on patients’ HRQOL (Swan et al., 2015) (lines 72-76). We did not mention the work by Matar et al. (2020), because it is a primary evidence on OD profiling in an acute care setting and not a literature review.

you need to be clear why you have focused on the common sense illness model – it comes out of the blue

As suggested by the Editor, in the introduction we provided the rationale for using the Common-Sense Model of Self-Regulation (CSM), namely the interpretation of needs-driven behavioral and psychological adaptation to OD. We subsequently described the model (lines 99-118). The CSM is grounded in the stress and coping theory (Lazarus, 2000) and in our opinion it represents an exhaustive framework for interpreting patients’ mental representation of a health condition, and its links with action plans and health outcomes. Other health models do exist, but they are aimed at explaining individuals’ involvement in prevention programs based on the role of individuals’ beliefs (Health Belief Model, Rosenstock, 1996), motivation (Protection Motivation Theory, Rogers, 1983) and intention (Theory of Planned Behavior, Azjen, 1988). Thus, the CSM seemed the most appropriate framework to interpret patients’ and caregivers’ experiences of OD and OD-related needs.

There are lots of aims from line 77, then research questions from line 82 then hypothesis from line 85 – please give one general aim or purpose as a focus at the end of the introduction giving a greater amount of space to the rationale needed above – else I think it is confusing for the reader

In the attempt to better clarify the aim of our scoping review, as suggested by the Editor, we expanded the review rationale, describing (a) the increased attention that was paid to patients’ subjective illness experiences since the introduction of the International Classification of Functioning, Disability and Health (ICF) by the World Health Organization, the Charter on Medical Professionalism and the SPIKES protocol (lines 137-145), (b) the existing OD-related literature reviews (lines 72-76), and (c) the existing models for the classification of patients’ and caregivers’ needs (lines 84-94). Besides, we formulated one general aim with one specific research question and related hypotheses, and removed all other redundant information (lines 120-136). The review aim reported in the Abstract was changed accordingly (lines 21-26).

Methods

Was there a protocol published for this review e.g., PROSPERO?

As recommended by Tricco et al. (2018), a scoping review protocol was agreed upon and shared by all team members. We acknowledge the importance of publishing the review protocol in order to enhance transparency in the review process and reduce unplanned review duplication. Nevertheless, it was not possible to publish our review protocol on PROSPERO, since this database does not accept scoping review protocols. We thus registered the review protocol and the data extraction form on Open Science Framework (https://osf.io/registries?view_only=). The registration is available at the following doi: 10.17605/OSF.IO/6ZCPK (line 158). 

I am not sure about the definitions placed here consider the introduction or a supplementary file. Can you use PICOS, SPIDER or another acronym to identify eligibility criteria first? Then this would give guidance for your key words also

As suggested, we integrated the operational definitions of oropharyngeal dysphagia, informal caregivers and needs in the Introduction section (respectively, lines 52-54, lines 68 and 69, and lines 78-83) and subsequently removed them from the Methods section (lines 169-185). We agree with the Editor that using a well-known framework such as PICO (Richardson et al., 1995) would increase clarity of reporting both the research question and the eligibility criteria. Therefore, we developed our research question using PICO, identifying adult persons with OD and/or their informal caregivers as Population, needs assessment as Intervention, and unmet or satisfied OD-related needs as Outcome. We deemed the parameter Comparison not applicable (lines 164-168). 

For the searching please just state when the final search took place and ignore the fact you searched earlier – please update the abstract also

As suggested by the Editor, in order to make the text clearer for the reader, we indicated the 17th February 2021 as the date of the final search in the Methods (line 194) and Abstract sections (line 30) and deleted a redundant sentence on lines 197 -199. The Results section (lines 260-262 and lines 267-273) and the PRISMA flow-diagram (Fig 1) were changed accordingly. As shown on the track changes version of the manuscript and Fig 1, 194 records were added to the “Records identified through database searching” box. As 30 duplicates had to be removed after the updated search, 164 records were added to the “Records after duplicates removed” box. Since all records but one were excluded after reading title and abstract, 163 records were added to the “Records excluded” box and one to the “Full-text articles assessed for eligibility” box. Moreover, two records were added to the “Additional records identified through other sources” box (see the first answer to the Reviewer’s comment on page 11 of the present letter). Finally, three records were added to the “Studies included in qualitative synthesis” box.

I found the eligibility criteria hard to follow – you identify constraints line 142-143 and then more specific criteria - can you just have one place where you identify criteria and split criteria into inclusion and exclusion – please do not have the opposite criteria for exclusion just other criteria

Give rationale for your criteria e.g., why after 2000? Why did studies have to involve less than 10 participants?

In order to clarify the text for the reader, we restructured inclusion and exclusion criteria in one paragraph. We defined inclusion criteria based on PICO and described all the other constraints as exclusion criteria (lines 208-215 and lines 220 and 221). We deleted the fourth and fifth exclusion criteria (i.e. “the clinical condition considered was esophageal dysphagia” and “the study consisted in a trial protocol”), as they mirrored the opposite of inclusion criteria (lines 219 and 220). We rephrased the rationale for exclusion criteria (lines 222-225). In particular, constraints regarding language and year of publication ensured the feasibility of full-text analysis and the inclusion of updated evidence, respectively. Studies including a very low number of participants (<10), lacking meaningful information and not subjected to peer-review were excluded, in order to guarantee the highest possible methodological standards.

As you are dealing with a lot of qualitative studies can you provide a methodology and paradigmatic position upfront for your review

We agree with the Editor on the importance of reporting a paradigmatic position when performing a review of qualitative research studies. Thus, on lines 153-156 we made explicit reference to the “best fit” Framework synthesis approach (Dixon-Woods, 2011; Carrol et al., 2011), a mainly deductive approach which is however integrated with an inductive perspective. Adopting this methodology, we were able to use the Supportive Care Framework (Fitch, 2008) as a scaffold, at the same time enriching and broadening the results of our review with the voices of participants from included studies.

The reader will need to understand how you came up with your results and an audit trail is needed in a supplementary file to show stages of analysis

As detailed in the Methods section, relevant studies were identified through database search with a systematic search strategy (detailed in S2 Table) and screened using the software Rayyan.qcri (Ouzzani, 2016) with ad-hoc inclusion and exclusion criteria (lines 211-225). Results of this process are represented as PRISMA flow diagram in Fig 1. Relevant study characteristics (shown in Table 1) and verbatim quotations of OD-related needs were extracted using a data-charting form developed by the research team. For each extracted need, participants’ exemplary verbatim quotations and the wording of questionnaire items referring to OD-related needs were reported for qualitative and quantitative studies, respectively for each need, along with the authors’ analytic interpretations. The Supportive Care Framework (SCF; Fitch, 2008) was used as a scaffold against which to map the emerging themes. A preliminary need domain categorization was performed by two researchers and subsequently refined by adding lower classification levels, labelled as sub-categories. Sub-category definitions for each SCF domain are reported in Table 3, as per Editor’s suggestion. Finally, a critical appraisal of the eligible publications was performed through the Critical Appraisal Checklist for Qualitative Research tools developed by Joanna Briggs Institute (JBI; Lockwood et al., 2020). Results of the critical appraisal are reported as supplementary material (S3 Table).

The database search process, the characteristics of the included studies and the critical appraisal are described in Fig 1, Table 1, and S3 Table, respectively. Exemplary quotations, as well as absolute (relative) frequencies of the needs were reported for each category and sub-category (Table 3). As per reviewer suggestion, an audit trail is now added as supplementary file (S6 Table), in order to increase transparency and reproducibility of the coding process. We added the reference to the audit trail to the Results section (lines 318 and 319) and to the Supporting Information Captions (line 788).

The analysis will have to address integration between qualitative and quantitative data – as you have different designs of studies

We thank the Editor for highlighting the issue of quantitative and qualitative data integration and the need for explicit reference to the methods used for our review. On lines 237-239, we specified that for qualitative studies the unit of analysis was represented by participants’ verbatim quotations, while for the quantitative study we considered the wording of OD-related need questionnaire items (i.e. “To what extent did the “type of liquid” quench your thirst?” and “Are you satisfied with your fluids?”). Following the Joanna Briggs Institute guidelines for the conduction of mixed-methods systematic reviews (Lizarondo et al., 2020), since the research question of our review could be addressed through both qualitative and scaled answers, we adopted a convergent integrated approach. In more detail, the synthesis of the different types of answers occurred simultaneously, using qualitizing of quantitative data as process of data transformation, followed by the integration of findings through a “best fit” Framework synthesis approach (lines 242-247). 

How was critical appraisal used for different types of study and what impact does it have on the results. Please really consider this

As reported in the manuscript (lines 232 and 233) the Critical Appraisal Checklist for Qualitative Research tools developed by Joanna Briggs Institute (JBI) was used to appraise the quality of the included qualitative studies. The only quantitative study included in the review (Howard et al., 2018) was an observational retrospective study conducted on a single group of patients. The aim of this study was to investigate the potential therapeutic intervention of textured thin liquids on nutrition, swallowing, and patients’ satisfaction and to provide viscosity measures for textured liquids. For the purpose of our review, in Howard et al.’s study we considered patients’ experiences only, namely the satisfaction with textured thin liquids. Thus, we transformed quantitative data derived from questionnaires of Howard et al.’s study into qualitative data through a qualitizing process. Therefore, we chose to apply to Howard et al.’s study the Critical Appraisal Checklist for Qualitative Research tools developed by Joanna Briggs Institute (JBI).

As concerns the impact on review results, critical appraisal was not considered an exclusion criterion, rather it informed results interpretation, highlighting potential flaws and need for further investigations (lines 233-235). We additionally discussed the possible implications of an unsatisfactory critical appraisal on the review results on lines 394-399. In particular, in the present review methodological quality was satisfactory for all studies except one, in which authors’ conclusions were only partially supported by participants’ voices. Nevertheless, we considered as unit of analysis only participants’ verbatim quotations reported in the Results section of the included publications, and not authors’ discussion on their results. Thus, we argue that the unsatisfactory critical appraisal of this specific study would not impact the results of the present review. 

Results

Can you add some sub-headings for needs categorization

As per Editor’s suggestion, in order to make the presentation of results clearer and the methodology reproducible, we added the definitions used for need categorization to Table 3.

The reader needs to understand how study quality influenced results – especially for interventions

We thank the Editor for highlighting the importance of study quality, especially when examining interventions effectiveness. Nevertheless, we respectfully point out that our aim was to explore what is known in the literature about patients’ and caregivers’ OD-related needs. All the included publications demonstrated a satisfactory methodological quality, with the exception of one study, in which authors’ conclusions were only partially supported by participants’ voices. Only participants’ verbatim quotations reported in the Results section were considered as units of analysis; therefore, we argue that the unsatisfactory critical appraisal emerging in this specific case would not impact the results of the review. Consistent with the research purpose, we reported on the few intervention studies retrieved and the conclusions drawn by their authors. We believe that a careful consideration of the influence of study quality on the effectiveness of the interventions goes beyond the scope of the present review. In fact, it would be more appropriate for a systematic review on effectiveness of interventions for needs satisfaction. We added these considerations to the Discussion section on lines 394-399.

You need to consider the link to the common sense model of illness

We acknowledge the importance of organically integrating the Common-Sense Model of Self-Regulation (CSM) throughout the manuscript. In the attempt to do so, we moved the description of the CSM from the Discussion (lines 453-470) to the Introduction (lines 99-118). As reported in the Introduction (lines 99-102), the CSM would serve as a useful theoretical framework for results interpretation, while the Supportive Care Framework (SCF) for needs categorization. As such, it seemed more appropriate to refer to the SCF in the Results section and to the CSM in both the Introduction and Discussion sections, in order to propose an interpretation of the retrieved needs based on a solid pertinent theoretical framework. 

Discussion

The discussion lacks references can you check list and develop it further

The common sense illness model if defined here which is too late – does your data really show this model from anticipating threat to coping? for both patients and carers? I think this distracts the reader quite a lot from what was coming through the data - consider and justify in your response if, and how much time is allocated to this

As per Editor’s suggestion, we expanded the Discussion section adding references about intrinsic limitations of qualitative research designs (Leung, 2015; line 390), OD studies from Eastern Asian countries (Xia et al., 2011; Yang et al., 2016; line 404), and the relative importance of swallowing impairments for patients suffering from progressive neurologic conditions (Howells et al., 2021a; Howells et al., 2021b; line 411).

Additionally, we expanded the discussion elaborating on the nature of needs and the appropriate tools for their investigation (lines 424-438). We also discussed the application of the SCF to the results of the present review and provided reasons for introducing slight changes in the original model (lines 443-450).

As concerns the CSM, as previously clarified in this response letter, we moved its description and the rationale for its use to the Introduction section (lines 99-118). We believe the OD-related needs retrieved in the present review can be fruitfully interpreted through the lens of the CSM. In fact, needs are a subjective construct by definition. The CSM deals with the lay individual representations of illness and related treatment, that represent the starting point for planning and evaluating coping strategies, at the cognitive and emotional levels. Moreover, the CSM focuses on the coherence (explanatory meaning) that an illness has for the person. Thus, it is our modest opinion that a critical interpretation of the results retrieved in the literature through the lens of an existing and well-established theoretical framework could add value and contribute to the debate among researchers dealing with the same topic, rather than distracting the reader.

Reviewer #1

Thank you for the opportunity to review this manuscript. This is an important area of research that has received little attention to date. The manuscript is well written and the methods are clearly described. My main concern regarding the manuscript is that there appears to be a number of relevant studies that have not been included. Perhaps they have been excluded for valid reasons but this is not clear based on the inclusion criteria. Publications such as Patterson et al., 2013; Howells et al., 2020a and Howells et al., 2020b. If these studies have been excluded, it is suggested that the authors carefully consider the wording of their inclusion/exclusion criteria to further highlight why these types of studies have been excluded. This is to ensure that the study methodology is reproducible. If there are studies that have been missed, such as the above, then the results will need to be reanalysed and the discussion may need to be reworked depending on whether changes to the results are required.

We thank the Reviewer for highlighting the concerns on relevant excluded manuscripts. As explained in the response to the Editor’s comments, we revised inclusion and exclusion criteria based on PICO question (208-215 and lines 220 and 221), deleted the exclusion criteria that mirrored the opposite of inclusion criteria (lines 219 and 220), and rephrased the rationale for exclusion criteria (lines 222-225). 

Studies in which dysphagia etiology was unclearly described were excluded, since it was not possible to distinguish between esophageal and oropharyngeal dysphagia. In addition, as explained on lines 431-438, studies reporting on patients’ functional loss through questionnaires without assessing the value individually attributed to the lost function did not include sufficient information regarding needs. Conversely, qualitative methods (e.g. interviews and focus-groups) and questionnaires specifically targeting needs were considered as proper tools for needs assessment. Thus, many studies investigating HRQOL through questionnaires were excluded from the present review. This was one of the reasons for excluding Patterson et al.’s study (2013). The other reason was related to the low number (8) of the caregivers interviewed by the authors regarding the experience of caring for someone with dysphagia. In order to ensure the highest possible methodological standards, we deemed as eligible for the present review studies involving at least 10 participants. 

The studies by Howells et al. (2021a, 2021b) were not retrieved by the search performed prior to submission of the manuscript to PLOS One. Due to the recent publication of these articles, a possible explanation is that they had not been indexed yet at the time of the previous research. However, since they present very rich and detailed experiences of both OD patients’ and caregivers’ experiences, we included them in the revised version of the manuscript, as records identified through manual search. The Abstract, Results, Discussion sections and the supplementary materials S1, S3, S4, and S5 were revised in order to include the new results. Overall, the citations retrieved in Howells et al.’s articles confirmed the categories distribution of previous results with only slight changes.

In addition, it was noted that the quote provided in Table 3 - Information needs (page 17) reference 18 is incorrect as this quote is not from this article. It is suggested that the authors double check all included quotes and the respective studies.

We thank the Reviewer for carefully checking our manuscript and apologize for the mistake regarding reference 18. After careful checking of all the included quotes, we corrected Table 3 substituting the number reference 18 with 51.

---

## [Decision Letter · Decision Letter 1]

8 Sep 2021

The care needs of persons with oropharyngeal dysphagia and their informal caregivers: A scoping review

PONE-D-21-06132R1

Dear Dr. Ninfa,

We’re pleased to inform you that your manuscript has been judged scientifically suitable for publication and will be formally accepted for publication once it meets all outstanding technical requirements.

Kind regards,

Andrew Soundy

Academic Editor

PLOS ONE

Additional Editor Comments (optional):

Thank you for your positive responses. I wish you the very best for your important research going forward.

Reviewers' comments:

Reviewer's Responses to Questions

**Comments to the Author**

1. If the authors have adequately addressed your comments raised in a previous round of review and you feel that this manuscript is now acceptable for publication, you may indicate that here to bypass the “Comments to the Author” section, enter your conflict of interest statement in the “Confidential to Editor” section, and submit your "Accept" recommendation.

Reviewer #2: All comments have been addressed

2. Is the manuscript technically sound, and do the data support the conclusions?

Reviewer #2: Yes

3. Has the statistical analysis been performed appropriately and rigorously? 

Reviewer #2: Yes

4. Have the authors made all data underlying the findings in their manuscript fully available?

Reviewer #2: Yes

5. Is the manuscript presented in an intelligible fashion and written in standard English?

Reviewer #2: Yes

6. Review Comments to the Author

Reviewer #2: I would like to congratulate the topic addressed, since it is not always an easy area to deal with. The Submission has been greatly improved and is worthy of publication.

7. PLOS authors have the option to publish the peer review history of their article (what does this mean?). If published, this will include your full peer review and any attached files.

Reviewer #2: No

---

## [Editor Report · Acceptance letter]

15 Sep 2021

PONE-D-21-06132R1 

The care needs of persons with oropharyngeal dysphagia and their informal caregivers: A scoping review 

Dear Dr. Ninfa:

I'm pleased to inform you that your manuscript has been deemed suitable for publication in PLOS ONE. Congratulations! Your manuscript is now with our production department. 

Kind regards, 

on behalf of

Dr. Andrew Soundy 

Academic Editor

PLOS ONE